# The Impact of the Dialysis Event Prevention Bundle on the Reduction in Dialysis Event Rate in Patients with Catheters: A Retrospective and Prospective Cohort Study

**DOI:** 10.3390/diseases12120301

**Published:** 2024-11-24

**Authors:** Reem Hamed AlHulays, Amany A. Ghazy, Ahmed E. Taha

**Affiliations:** 1Prince Mansour Military Hospital, Taif 26315, Saudi Arabia; reemalhulays@gmail.com; 2Medical Microbiology and Immunology Unit, Department of Pathology, College of Medicine, Jouf University, Sakaka 72388, Saudi Arabia; aeattia@ju.edu.sa; 3Medical Microbiology and Immunology Department, Faculty of Medicine, Mansoura University, Mansoura 35516, Egypt

**Keywords:** dialysis event, dialysis event prevention bundle, hemodialysis, CRBSI

## Abstract

**Background:** Dialysis-associated events such as bloodstream infections represent serious complications for hemodialysis patients, with the potential to increase morbidity and mortality. **Aims:** To assess the impact of implementing a comprehensive bundle of evidence-based practice on reducing dialysis event rates among catheter dialysis patients at Prince Mansour Military Hospital Dialysis Center. **Participants and Methods:** The study enrolled 111 hemodialysis participants. A comprehensive dialysis event prevention bundle consisting of 6 key components was implemented. **Results:** Implementation of the dialysis event prevention bundle showed a significant decrease in IV antimicrobial start (*p* = 0.003), positive blood culture (*p* = 0.039), and inflammation at the vascular access site eliminated (*p* = 0.004). There was a positive correlation between IV antimicrobial start and both patients’ age (*p* = 0.005) and the permanent catheter site (*p* = 0.002). Positive blood culture was significantly correlated with comorbidities (*p* = 0.000) and patients’ age (*p* = 0.320). A positive correlation between pus, redness, or increased swelling at the vascular access site with comorbidities (*p* = 0.034), patients’ age (*p* = 0.021), and the permanent catheter site (*p* = 0.002) was observed. Staff compliance with the dialysis event prevention bundle components has improved regarding hemodialysis catheter disconnection, catheter exit site care, and routine disinfection. **Conclusions:** Implementation of a comprehensive dialysis event prevention bundle can effectively reduce dialysis event rates and enhance patient safety.

## 1. Background

Hemodialysis is a therapy that maintains the patient’s life for most end-stage kidney sufferers. However, many hazards are associated with hemodialysis, like bloodstream infections related to catheters, issues with vascular access, and fluid imbalances [1].

Hemodialysis patients are at high risk of infection because of their weakened cellular immunity, neutrophil function, and complement activation, in addition to the presence of uremia, and frequent access to their bloodstream, especially those with non-tunneled and tunneled catheters [2]. Thus, the type of vascular access used during their dialysis session, whether it is a catheter, fistula, or graft, can significantly impact their health. Bloodstream infections and localized infections at the vascular access site are not just complications, but they can lead to morbidity and mortality in dialysis patients. The use of a catheter in hemodialysis, for instance, can escalate the risk of bloodstream infection and localized vascular access infections. Furthermore, hemodialysis patients are at a heightened risk of infection with multi-drug-resistant bacteria [3].

In Saudi Arabia, end-stage renal illness is a high-alert concern. As of the most recent update, over 20,000 individuals were on dialysis, and 9810 patients were receiving follow-up care following a kidney transplant [1].

The Center for Disease Control and Prevention (CDC) has declared that the dialysis event should be recorded when any conditions met positive blood culture, localized redness, swelling, or pus at dialysis access, or initiation of an intravenous antimicrobial drug [4].

In response to the importance of promoting patient safety and reducing the risks associated with dialysis, healthcare organizations, the CDC and the Saudi Ministry of Health (MOH) have developed highly effective comprehensive interventions known as dialysis event prevention bundles. The dialysis event prevention bundle is a set of evidence-based, systematic practices adopted by the CDC and MOH to reduce unnecessary clinical variation, prevent dialysis-related events, and provide high-quality services [5,6].

Dialysis event prevention bundles encompass various components such as infection control measurements in hemodialysis catheter connection, hemodialysis catheter disconnection, hemodialysis catheter exit site care, dialysis station routine disinfection, hemodialysis injectable medication preparation, and hemodialysis injectable medication administration. Figure 1 lists the six components of dialysis event prevention bundles [5,6].

Researchers have been interested in the impact of these bundles on reducing dialysis event rates, particularly catheter-related bloodstream infections (CRBSI). CRBSIs mean the presence of bacteremia due to an intravenous catheter. They are emerging causes of hospital-acquired infection that are associated with high morbidity, mortality, and cost [7]. A multicenter, stepped wedge, cluster-randomized controlled trial study conducted by Weikert et al. (2024) was conducted to investigate the effect of a multimodal prevention strategy on dialysis-associated infection events among 11,251 hemodialysis patients in 43 hemodialysis outpatient facilities with 1,413,457 proceeded hemodialysis. They noticed that incidence rates of dialysis-associated infection have decreased from 0.71 to 0.31 per 1000 dialyses in the intervention group [8].

In Saudi Arabia, infection prevention and control (IPC) measures are considered vital targets for the MOH in seeking patient, visitor, and staff safety. In 2024, the MOH declared the manual of the General Directorate of Infection Prevention and Control of Healthcare Facilities (GDIPC) as an essential tool to assess the proper implementation of IPC measures in clinical facilities. This manual serves as a guide for the infection control audit in a standardized methodology to ensure the proper application of IPC measures at the highest level of quality [9].

Based on the current literature review across various databases, the current study is considered the first nationally to assess the impact of implementing a comprehensive bundle of evidence-based practice on reducing dialysis event rates among catheter dialysis patients at Prince Mansour Military Hospital Dialysis Center in the Taif Region and measuring staff compliance with the newly introduced dialysis event prevention bundle. It ultimately aims to improve patient safety and enhance the quality of care provided during dialysis sessions.

## 2. Subjects and Methods

### 2.1. Study Setting and Design

The study was conducted at the Prince Mansour Military Hospital Dialysis Center, a facility located in the Taif Region. This center, equipped with 40 beds, serves approximately 189 dialysis patients.

This study employed retrospective and prospective study design. The retrospective side of the study (pre-implementation phase) covered six months before the implementation of the dialysis prevention bundle, starting in November 2023 and ending in April 2024. In the pre-implementation phase, the dialysis event rates were monitored for all the patients who met the eligibility criteria followed by the prospective side of the study (implementation phase), covered two months after executing the bioethical approval in May 2024.

Patients with catheters undergoing dialysis were meticulously monitored to assess the impact of the dialysis event prevention bundle applied at the time of connection, disconnection, catheter site care, dialysis station disinfection, and hemodialysis injectable medication preparation and administration in reducing dialysis event rates.

### 2.2. Ethics Approval and Informed Consent

The ethics committee of Armed Forces Hospitals approved the study (H-02-T-078); approval number (REC. 2024-865). Informed consents were signed by patients enrolled in the study, and access to patient records was restricted to authorized personnel.

### 2.3. Participant Selection

A total of 111 participants were enrolled in the study based on specific (inclusion) criteria: outpatients with end-stage renal disease with tunneled and non-tunneled permanent catheters. Patients with both an implanted access (graft or fistula) and a catheter were included as catheter patients. Exclusion criteria include mono-hemodialysis (peritoneal dialysis), non-tunneled (temporary) catheters, chronic hemodialysis with implanted access (graft or fistula), and inpatients.

Initially, at the beginning of the study, the number of participants included was 123 patients; 12 participants were removed. The reasons for their removal were as follows: deceased, two participants passed away during the study period; hospital admission, two participants were admitted to the hospital for medical reasons unrelated to the study; dialysis sessions, four participants received dialysis treatment at other facilities; vacation, four participants were on a vacation and received dialysis treatment at other facilities outside the kingdom. As a result, the final sample size was 111 participants.

### 2.4. Study Interventions and Implementation

To decrease the dialysis events rate, a comprehensive dialysis event prevention bundle consisting of six key components: hemodialysis catheter connection, hemodialysis catheter disconnection, hemodialysis catheter exits site care, dialysis station routine disinfection, hemodialysis injectable medication preparation, and hemodialysis injectable medication administration was implemented (Figure 1). These components align with the Center for Disease Control and Prevention [5,6].

Before implementation, intensive teaching and training sessions were conducted. These sessions focused on essential topics such as the Hand Hygiene (WHO 5 Moments) protocol, which emphasizes the critical moments for hand hygiene to prevent healthcare-associated infections, aseptic techniques, cleaning, and disinfection methods, proper use of disinfectant agents, and catheter site care, including the selection of appropriate skin antiseptics [10]. Education materials were provided to both patients and healthcare workers.

Active surveillance of dialysis event rates was performed, with timely feedback reports that included details on positive blood cultures, bacterial types (Gram-positive or Gram-negative), and staff compliance with the dialysis event prevention bundle. The effectiveness of this approach was outlined in the literature for the core components of effective infection prevention and control programs: new WHO evidence-based recommendations [11]. The validity of the dialysis event prevention bundle was assessed by infection control practitioners and assigned infection control link nurses using the dialysis care bundle checklist. Ensuring the validity of infection control bundles requires a continuous assessment of their implementation and effectiveness [12].

Additionally, a committee comprising members from the infection control department, the dialysis center head, the head nurse, patient education, and the housekeeping supervisor was established. This committee conducted weekly visits to evaluate cleanliness, patient satisfaction, and awareness of infection control measures. Posters were distributed throughout the area to remind patients of their right to receive safe care and to encourage them to request hand hygiene compliance from healthcare workers.

### 2.5. Outcomes

A comprehensive baseline data collection involving patient demographics, comorbidities, catheter site, and dialysis event rate according to the CDC definition [4] was meticulously gathered before implementing the dialysis event prevention bundle. These data were collected by reviewing the patient medical record system to assess the doctor and nurse notes, communication log book, and laboratory system. Post-implementation data were collected prospectively following the implementation of the dialysis event prevention bundle. This includes dialysis event rates, such as any positive blood culture, redness, or swelling at the access site, starting any intravenous antimicrobial drug, and staff compliance with the newly introduced bundle.

### 2.6. Statistical Analyses

The data were analyzed using SPSS Software (version 26). The dialysis event rate before and after the intervention was compared using the independent sample *t*-test. Qualitative data were expressed by number (N.) and percent (%). Quantitative data were expressed by mean ± SD. The chi-square test was used to evaluate the relation between the studied parameters. *p*-values were measured, and the level of significance was <0.05.

## 3. Results

### 3.1. Patients’ Demographic and Clinical Data

Data were collected from 123 end-stage renal disease (ESRD) outpatients for all the study period months. A total of 12 were removed from the study, so the final number of participants included in the study was 111 patients. Across the study population of 111 patients, participants had a mean age of 55 years (standard deviation 17.38), 57 (51.0%) were male, and 88 (80%) had comorbidities, wherein 5 patients had diabetes mellitus (DM), 34 had hypertension (HTN), and 49 suffered from both DM and HTN. As for catheter characteristics, 95% of patients had tunneled intra-jugular permanent catheters (Table 1).

### 3.2. Prevalence of Gram-Positive and Gram-Negative Bacteria in Blood-Positive Cultures

The prevalence of Gram-positive bacteria was higher than that of Gram-negative bacteria in both the pre-and post-implementation periods. Specifically, Gram-positive bacteria constituted 78% of cultures in the pre-implementation period and 67% in the post-implementation period (Figure 2).

The distribution of bacteria observed in the study is summarized in Table 2. Results revealed that *Staphylococcus epidermidis* was the most prevalent bacteria, accounting for 52.20% of the total bacterial samples followed by *Staphylococcus hominis*, representing 17.40% of the samples (Table 2).

### 3.3. The Differences Between Pre-Implementation and Post-Implementation Bacterial Prevalence

An independent *t*-test was used to study the differences between pre-implementation and post-implementation regarding the prevalence of Gram-positive and Gram-negative bacteria. No significant difference was found regarding Gram-positive bacteria (*p* = 0.094, T-value = 1.987), or Gram-negative bacteria (*p* = 0.604, T-value = 0.548).

### 3.4. Dialysis Events Rate for Pre- and Post-Implementation Period

The implementation of the dialysis event prevention bundle showed a significant decrease in dialysis events by 51% (Figure 3). An independent *t*-test was used to study the differences between pre-implementation and post-implementation in dialysis event rates. The results showed a significant reduction in IV antimicrobial start events (*p*-value 0.003, T-value 1.295), the positive blood culture event rate significantly decreased (*p* = 0.039, T-value 2.031), and the pus, redness, or increased swelling at the vascular access site eliminated (*p*-value 0.004, T-value 1.048) (Table 3).

### 3.5. Relation Between Dialysis-Related Events and (Patients, Age Category, Permanent Catheter Site)

To study the relationship between dialysis-related events and (comorbidities, age, and permanent catheter site) a chi-square test was used, and the results are shown in Table 4. There was a positive correlation between IV antimicrobial start and both patients’ age (*p* = 0.005) and permanent catheter site (*p* = 0.002) while no relation was found with the comorbidities (DM, HTN, or both) (*p* = 0.736). Positive blood culture was significantly correlated with comorbidities (*p* = 0.000) and patients’ age (*p* = 0.320) while no relation was found with the permanent catheter site (*p* = 0.078). Furthermore, there was a significant positive correlation between pus, redness, or increased swelling at the vascular access site and comorbidities (DM, HTN, or both) (*p* = 0.034) and patients’ age (*p* = 0.021) and the permanent catheter site (*p* = 0.002) (Table 4).

### 3.6. Dialysis Event Prevention Bundle Compliance

Comparing staff compliance with the dialysis event prevention bundle components between May 2024 and June 2024, a slight decrease in compliance was observed for some areas such as hemodialysis catheter connection, which fell from 89% to 86%, hemodialysis injectable medication preparation decreased from 92% to 90%, and injectable medication administration declined from 95% to 93%. In contrast, compliance has improved in other areas such as hemodialysis catheter disconnection, which increased from 90% to 93%, catheter exit site care rose from 90% to 91%, and routine dialysis station disinfection improved from 95% to 96% (Figure 4, Table 5). However, these results were statistically non-significant (*p* = 0.07). A score of zero in the main component was documented if any of its sub-elements were not completed.

## 4. Discussion

Dialysis events represent serious complications for hemodialysis patients, with the potential to lead to increased morbidity and mortality. These patients are at high risk of infection because of their weakened immune systems and frequent access to their bloodstream, especially those with non-tunneled and tunneled catheters [2]. The objectives of implementing a dialysis event prevention bundle are to provide systematic, standardized, and evidence-based care to prevent or decrease dialysis-related events that threaten patient safety and provide high-quality services [5,6].

The current study aimed to assess the impact of implementing a comprehensive bundle of evidence-based practice on reducing dialysis event rates among catheter dialysis patients at Prince Mansour Military Hospital Dialysis Center. The results clarified that the Gram-positive bacteria constituted 76% of the bacterial cultures from blood samples, and 52.20% were staphylococcus epidermis which is consistent with findings from Suzuki et al. (2016) [13]. Their meta-analysis on bacteremia in hemodialysis patients concluded that 50% to 75% of the causative organisms are Gram-positive bacteria, while the remaining are Gram-negative. Among the Gram-positive organisms, *Staphylococcus aureus*, including *Methicillin-resistant Staphylococcus aureus* (MRSA), is the most common, followed by *Staphylococcus epidermidis* [13].

In accordance with our results, some researchers have reported that about two-thirds of catheters were infected with Gram-positive bacteria (mostly *Staphylococcus aureus* or *Staphylococcus epidermidis*) [14]. Others reported that about 61% of positive blood cultures were Gram-positive cocci (63% of them were Staphylococcal species and 63% of them were methicillin resistant) [15]. Also, in 2023, Almenara-Tejederas and his colleagues evaluated 406 tunneled catheters implanted in 325 haemodialysis patients and found that the incidence of catheter-related bacteremia was 0.40/1000 catheter days (mostly after 6 months of implantation) and the predominant bacteria were *Staphylococcus epidermidis* (48.4%) and *Staphylococcus aureus* (28.0%) [16].

The predominance of Gram-positive bacteria, particularly *Staphylococcus epidermidis*, can be attributed to inadequate infection control practices. Issues such as improper techniques during connection and disconnection of dialysis lines, insufficient precautions in injectable medication preparation and administration, and inadequate catheter site care, including skin antiseptics, aseptic techniques, and hand hygiene, can facilitate the entry of normal skin flora into the bloodstream.

Furthermore, the current study showed that the dialysis events rate can be reduced in the outpatient hemodialysis setting for ESRD patients with permanent catheter access by implementing a comprehensive dialysis prevention bundle, combined with intensive teaching and training sessions, an active infection control surveillance and feedback system, and patients’ involvement. Results demonstrated a significant reduction in dialysis events, with an overall decrease of 51%. Specifically, the rate of IV antimicrobial start events was reduced by 45.5% (*p*-value 0.003) indicating a substantial impact. The t-value for this event was 1.295, which suggests a moderate effect size relative to the variability in the data. The positive blood culture events showed a 55% reduction. This reduction was statistically significant, with a *p*-value of 0.039. The t-value for this measure was 2.031, indicating a more substantial effect size compared to IV antimicrobial start. The pus, redness, or increased swelling at the vascular access site has achieved a remarkable 100% reduction in this event (*p* = 0.004). These findings align with those of Gork et al. (2019) who implemented similar dialysis event prevention bundles intending to decrease dialysis event rates. Their study achieved a notable 70% reduction in dialysis event rates through the use of a checklist bundle and surveillance program [17].

However, the only meta-analysis systematic review relevant to this topic was conducted by Lavallée et al. (2017) who assessed the impact of care bundles on patient outcomes more generally and found the evidence for their effectiveness to be of low quality. Notably, this meta-analysis did not include the dialysis event prevention bundle [18].

Kotwal et al. (2020) have conducted a stepped wedge, cluster-randomized trial across 37 renal services in Australia, involving 6364 patients, to evaluate multifaceted interventions or care bundles to reduce catheter-related bloodstream infections (CRBSIs). The study concluded that implementing a multifaceted intervention has no impact on reducing CRBSI. The researchers noted limitations such as the incomplete recording of all catheters and infection events from every service and the lack of active feedback on infection data and hand hygiene compliance, which may have affected the results [19].

There is only one study by Gork et al. (2019) that applied dialysis event prevention bundles similar to the current bundle; the goal of the study is to decrease the rate of dialysis events by implementing an intervention (checklist bundle) and surveillance program. The study results show a significant decrease in the dialysis events rate by 70% [17].

Some studies have evaluated the use of other medications rather than antibiotics to reduce catheter exit site infection [20,21]. Johnsons et al. (2005) have determined the efficacy and safety of using Medihoney versus mupirocin in preventing catheter exit site-associated infections among 101 hemodialysis patients. They noticed that incidences of bacterial infections were comparable in both groups (0.97 vs. 0.85 episodes per 1000 catheter days) and 2% of staphylococcal isolates were mupirocin resistant. Honey was effective against antibiotic-resistant microorganisms [20]. Power et al. (2009) have evaluated the long-term use of 46.7% citrate catheter locks and found no significant difference in rates of catheter exit site infection (0.7 vs. 0.5 events/1000 catheter-days [21].

On the other hand, a multicenter, stepped wedge, cluster-randomized controlled trial applied the same type of study to evaluate the effect of a multimodal prevention strategy on dialysis-associated infection events among hemodialysis outpatients. The result showed an overall risk reduction of more than 50% after utilizing the prevention strategy [8].

Regarding the staff compliance with the dialysis event prevention bundle, results clarified a slight decrease in compliance was observed for some areas such as hemodialysis catheter connection (from 89% to 86%), hemodialysis injectable medication preparation (from 92% to 90%), and injectable medication administration (from 95% to 93%). In contrast, compliance has improved in other areas such as hemodialysis catheter disconnection increased from 90% to 93%, catheter exit site care rose from 90% to 91%, and routine dialysis station disinfection improved from 95% to 96%. However, these results were statistically non-significant (*p* = 0.07). A score of zero in the main component was documented if any of its sub-elements were not completed. Several reasons exist for the slight decrease in staff compliance with the dialysis event prevention such as the assignment of six new nurses to the hemodialysis center without proper training and education. Additionally, no further training or education sessions were conducted in the second month after implementing the dialysis event prevention bundle. Although immediate feedback was provided during bundle compliance audits, and staff were educated on compliance issues, the need for ongoing training and integrating new, untrained staff likely impacted overall adherence.

The current study has several limitations. First, the implementation period was too short. For a comprehensive evaluation of a quality intervention, a more extended period is needed to assess its sustained impact. Second, the study’s sample size was limited. Approximately 20 patients with permanent catheters were excluded because their dialysis treatments occurred at other facilities within the current center’s network. Third, infection control practitioners audited and assessed staff compliance with the dialysis event prevention bundle. Due to staffing shortages in the IPC department, hemodialysis infection control link nurses were also involved in auditing. This addition may have introduced bias, as the link nurses might have influenced the audit results to reflect better compliance. Additionally, the infection control practitioners and link nurses were not always present in the ward. Furthermore, the infection control link nurse had other responsibilities in addition to bundle auditing, which may have also impacted the thoroughness of their evaluations. Fourth, the effect of each component of the dialysis event prevention bundle had to be evaluated separately. This limitation makes it difficult to determine the specific impact of each intervention component on the overall outcomes.

Based on the limitations identified in the current study, several areas warrant further investigation such as extended implementation period and component-specific analysis. In future studies, we will extend the implementation period to evaluate the long-term effect of the dialysis event prevention bundle on the dialysis event rate. A longer implementation duration will provide a clearer understanding of which elements of the dialysis event prevention bundle are the most effective. In addition to addressing the limitations of the current study, further research should focus on understanding and mitigating the predominance of Gram-positive bacteria in bloodstream infections among hemodialysis patients.

## 5. Conclusions

Implementation of a comprehensive dialysis event prevention bundle can effectively reduce dialysis event rates and enhance patient safety.

## Figures and Tables

**Figure 1 diseases-12-00301-f001:**
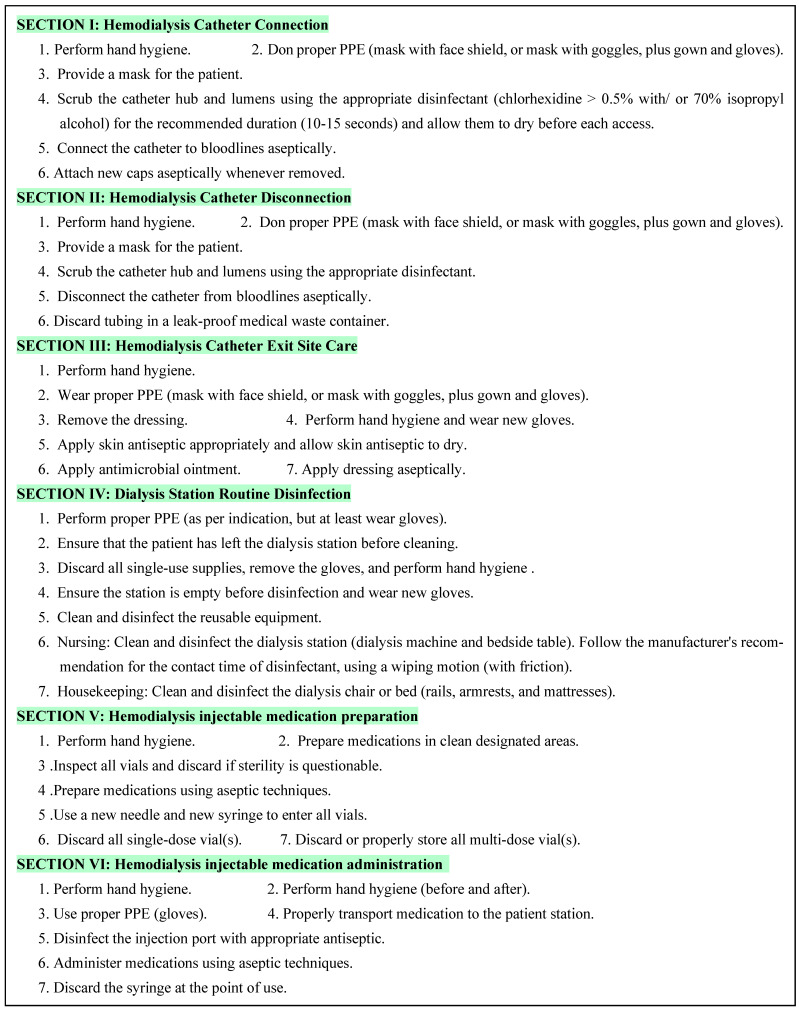
The six components of the dialysis event prevention bundles.

**Figure 2 diseases-12-00301-f002:**
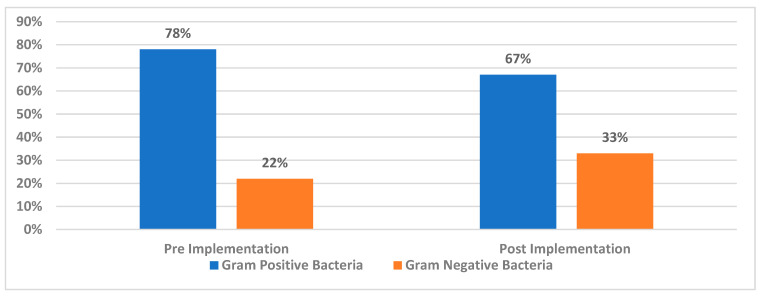
Percentage of Gram-negative and positive bacteria for pre- and post-implementation period.

**Figure 3 diseases-12-00301-f003:**
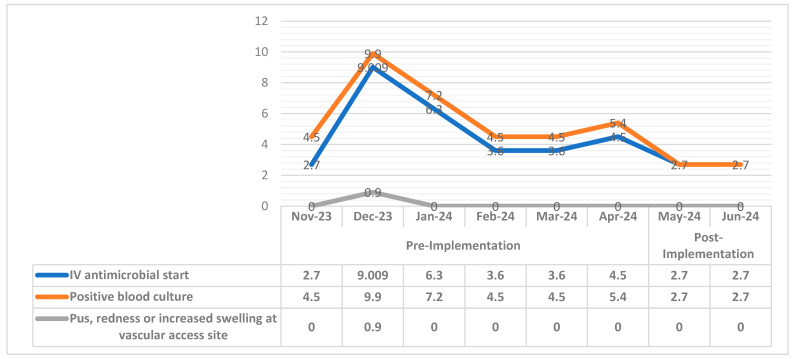
Dialysis events rate by month.

**Figure 4 diseases-12-00301-f004:**
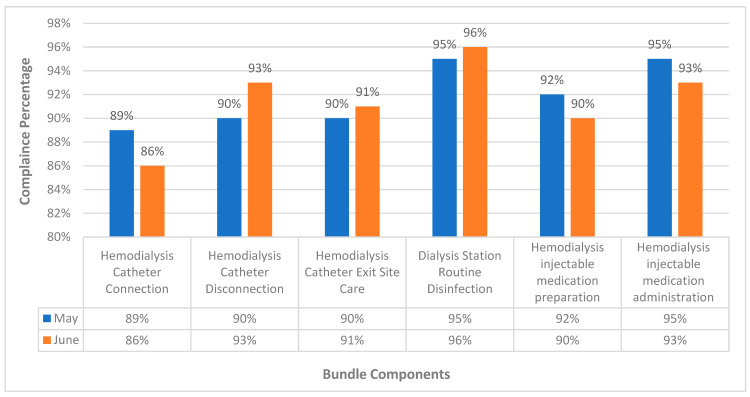
Percentage of staff compliance to dialysis event prevention bundle.

**Table 1 diseases-12-00301-t001:** Patients’ demographic and clinical data.

Category	Number (N.)	Percent (%)
**Gender**		
Males	57	51%
Female	54	49%
**Age Category:**		
18–20 years	2	1.8%
20–40 years	22	19.8%
41–60 years	43	38.6%
61–90 years	44	39.6%
**Comorbidities**		
Patients With Comorbidities (DM, HTN, or both)	88	79.3%
Patients Without Comorbidities	23	21.6%
**Permanent Catheter Site:**		
Jugular Permanent Catheter (JPC)	106	95.5%
Femoral Permanent Catheter (FPC)	4	3.6%
Trans-hepatic Permanent Catheter (THPC)	1	0.9%

DM: diabetes mellitus; HTN: hypertension.

**Table 2 diseases-12-00301-t002:** Bacteria distribution by prevalence per month.

Bacteria	Pre-Implementation						Post-Implementation		Total	Percentage
	November-23	December-23	January-24	February-24	March-24	April-24	May-24	June-24		
**Gram-Positive Bacteria**										
*Staphylococcus epidermidis*	2	6	4	1	3	6	0	2	24	52.20%
*Staphylococcus hominis*	2	3	0	1	1	0	1	0	8	17.40%
*Staphylococcus haemolyticus*	1	0	0	0	1	0	0	0	2	4.30%
*Staphylococcus lugdunensis*	0	0	1	0	0	0	0	0	1	2.17%
*Staphylococcus capitis*	0	0	0	0	0	0	1	0	1	2.17%
**Gram Negative Bacteria**										
*Klebsiella oxytoca*	0	1	0	0	0	0	0	0	1	2.17%
*Citrobacter koseri*	0	1	0	0	0	0	0	0	1	2.17%
*Pseudomonas aeruginosa*	0	0	1	1	0	0	0	1	3	6.52%
*Eschrichia coli*	0	0	1	0	0	0	0	0	1	2.17%
*Pantoea species*	0	0	1	0	0	0	0	0	1	2.17%
*Stenotrophomonas maltophilia*	0	0	0	1	0	0	0	0	1	2.17%
*Enterobacter cloacae*	0	0	0	1	0	0	0	0	1	2.17%
*Comamonas acidovorans*	0	0	0	0	0	0	1	0	1	2.17%

**Table 3 diseases-12-00301-t003:** Differences between pre-implementation and post-implementation in dialysis event rate.

Dialysis Event Rate	Group	N	Mean	Std. Deviation	*p*-Value	T
IV antimicrobial start	Pre-Implementation	6	4.95150	2.332722	0.003	1.295
	Post-Implementation	2	2.70000	0.000000		
Positive blood culture	Pre-Implementation	6	6.00000	2.179908	0.039	2.031
	Post-Implementation	2	2.70000	0.000000		
Pus, redness or increased swelling at vascular access site	Pre-Implementation	6	0.15000	0.367423	0.004	1.048
	Post-Implementation	2	0.00000	0.000000		

**Table 4 diseases-12-00301-t004:** Relations between dialysis-related events and (comorbidities, age, permanent catheter site).

**(1) Relation Between IV Antimicrobial Start and (Comorbidities, Age, Permanent Catheter Site)**	**No.**	**%**	** *p* ** **-Value**	**Chi-Square**
**Comorbidities**				
Patients With Comorbidities (DM, HTN, or both)	88	79.3%	0.736	1.982
Patients Without Comorbidities	23	21.6%		
**Age Category:**				
18–20 years	2	1.8%	0.005	4.173
20–40 years	22	19.8%		
41–60 years	43	38.7%		
61–90 years	44	39.6%		
**Permanent Catheter Site:**				
Jugular **Permanent** Catheter (JPC)	106	95.5%	0.002	4.904
Femoral **Permanent** Catheter (FPC)	4	3.6%		
Trans-hepatic **Permanent** Catheter (THPC)	1	0.9%		
**(2) Relation between Positive blood culture and (** **Comorbidities, age, permanent catheter site)**	**No.**	**%**	** *p* ** **-value**	**Chi-Square**
**Comorbidities**				
Patients With Comorbidities (DM, HTN, or both)	88	79.3%	0.000	5.012
Patients Without Comorbidities	23	21.6%		
**Age Category:**				
18–20 years	2	1.8%	0.320	1.723
20–40 years	22	19.8%		
41–60 years	43	38.7%		
61–90 years	44	39.6%		
**Permanent Catheter Site:**				
Jugular Permanent Catheter (JPC)	106	95.5%	0.078	1.884
Femoral Permanent Catheter (FPC)	4	3.6%		
Trans-hepatic Permanent Catheter (THPC)	1	0.9%		
**(3) Relation between pus, redness, or increased swelling at vascular access site and (** **Comorbidities, age, permanent catheter site)**	**No.**	**%**	** *p* ** **-value**	**Chi-Square**
**Comorbidities**				
Patients With Comorbidities (DM, HTN, or both)	88	79.3%	0.034	4.50
Patients Without Comorbidities	23	21.6%		
**Age Category:**				
18–20 years	2	1.8%	0.021	4.97
20–40 years	22	19.8%		
41–60 years	43	38.7%		
61–90 years	44	39.6%		
**Permanent Catheter Site:**				
Jugular Permanent Catheter (JPC)	106	95.5%	0.002	3.98
Femoral Permanent Catheter (FPC)	4	3.6%		
Trans-hepatic Permanent Catheter (THPC)	1	0.9%		

**Table 5 diseases-12-00301-t005:** Total number of dialysis event prevention bundle opportunities and staff compliance.

Dialysis Event Prevention Bundle’s Main Components	Total Opportunities (May 2024)	Staff Compliance (May 2024)	Total Opportunities (June 2024)	Staff Compliance (June 2024)
Hemodialysis catheter connection	1332	1185	1332	1150
Hemodialysis catheter disconnection	1332	1200	1332	1215
Hemodialysis catheter exit site care	1332	1200	1332	1250
Dialysis station routine disinfection	1332	1260	1332	1280
Hemodialysis injectable medication preparation	1332	1230	1332	1200
Hemodialysis injectable medication administration	1332	1265	1332	1240

## Data Availability

All data are available in the manuscript.

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
