# Peer review of "The Impact of the Dialysis Event Prevention Bundle on the Reduction in Dialysis Event Rate in Patients with Catheters: A Retrospective and Prospective Cohort Study"

_diseases, 2024, doi:10.3390/diseases12120301_

Round 1
Reviewer 1 Report
Comments and Suggestions for Authors
Dear authors
The article’s idea is not new, and as you stated in the discussions, several studies evaluated the implementation of specific actions to reduce the risk of dialysis-associated infections. Before publishing this article, several issues need to be addressed:
1. Please, present the prevention bundles as a supplementary material, or move them in the methods section.
Did you include in the study non-tunneled (temporary) catheters also. It is not clear.
2. In the statistical analysis and results section
Please evaluate the distribution of the continuous variables. As seen in table 1, age present a wide distribution with patients age raging between 18 and 90 years.
You said that 80% percent comorbidities. What comorbidities?
Please expand the statistical analysis section.
Please remove Figure 2. There is no point in including it in the paper. ( at least in my opinion )
Reformulate the sentence : “To study the significant differences between Pre-Implementation and Post-Implementation in the prevalence of Gram-Positive and Gram-Negative Bacteria, an independent sample T-test was used and indicated that there is no significant difference between Pre-Implementation and Post-Implementation in Gram-Positive Bacteria (P=0.094, T-value=1.987), with same conclusion to Gram-Negative Bacteria (P=0.604, T-value=0.548).”
I do not understand the section 3.4. Do you report the events per 1000 dialysis sessions? The idea is that your study has 6 months of pre-implementation and only two in post-implementation. It is logical that the number of events in 2 months is lower compared to 6 months. Again, please reread the phrases where you state that a p value higher than 0.05 is statistically significant!
I do not understand tables 4 and 5. You performed a chi-square test, but you should present the percentages for both headers, for instance with and without antimicrobial start, or with and without positive blood culture.
In the discussion section, please explain the results from the section 3.6 properly. Perhaps a cause for the reduced injectable administration represents the reduction in the infections, thus fewer antibiotics administered.
In the discussion section, please explain the results from the section 3.6 properly. Perhaps a cause for the reduced injectable administration represents the reduction in the infections, thus fewer antibiotics administered.
Please correct the double numbering of references
Good luck!
Comments on the Quality of English Language
Please correct the "medical english" and some typos (here and there)
Author Response
Dear Editor and reviewers,
I am very thankful to the reviewers for their deep and thorough review. I have revised my present research paper considering their useful suggestions and comments. I hope my revision has improved the paper to their satisfaction. Number-wise answers to their specific comments/suggestions/queries are in the attached file.
Please see the attachment.

Reviewer 2 Report
Comments and Suggestions for Authors
To the Editor of Diseases
It was with interest that I read Reem Hamed AlHulays et al.’s manuscript “Impact of Dialysis Event Prevention Bundle in The Reduction of Dialysis Event Rate in Patients with Catheters Retrospective and Prospective Cohort Study”.
COMMENTS:
The authors describe their single-center experience, through the implementation of the dialysis event prevention bundle in the reduction of dialysis event rate in patients with Central Venous Catheter (CVC) in a relative small cohort of participants (110 patients), with a mean age uniformly distributed (39% between 41-60 years and another 40% between 61-90). The implementation of the Dialysis Event Prevention Bundle showed a significant decrease in dialysis events by 51%, an important result. The manuscript is interesting and well written and debate about a topic that is always very much felt among clinicians.
Here some observations:
Maintenance chronic kidney disease (CKD) patients on hemodialysis treatment are at higher risk for infection, because uremia make patients with CKD more susceptible to infectious agents through defects in cellular immunity, neutrophil function and complement activation.
1- the incident rate of CVC infections would be better described as number of episodes per 1000 catheter days (Johnson, D.W. et al. Randomized, controlled trial of topical exit-site application of honey (Medihoney) versus mupirocin for the prevention of catheter-associated infections in hemodialysis patients. J Am Soc Nephrol. 2005; Power, A. et al. Sodium citrate versus heparin catheter locks for cuffed central venous catheters: a single-center randomized controlled trial. Am J Kidney Dis. 2009)
2- There is not a control group
3- The authors describe that 95% of patients had tunneled intra-jugular permanent catheters but do not itemize if it was on the right side, with a shorter route or on the left site, with a longer one?
4- The type of catheter was similar in all the patients?
5- The type of external disinfection was similar in all the patients? it would seem so, and if not, it has to be described because superior antisepsis seems to be better with chlorhexidine 2% compared with alcohol 70% or povidone-iodine 10% solution.
6- The prevalence of Gram-positive bacteria was the most important. Staphilococcus above all, accounting for 78% of gram + infectious agents and, fortunately, among Gram negative, only 6.5% of Pseudomonas Aeruginosa.
However, Staphilococcus aureus reported to be the most common agent of CVC infections, (Preventing Bloodstream Infections in People on Dialysis, CDC) updated on February 6th, 2023) was not found by the authors. In fact, more than 14,000 bloodstream infections are reported to occur in patients on dialysis in the U.S. in 2020, and more than one in three were caused by staphylococcus and staphylococcus aureus accounted for between 21 and 43% in most series, and methicillin-resistant S. aureus (MRSA) reported in approximately 12–38%
(Tanriover B et al. Bacteremia associated with tunneled dialysis catheters: comparison of two treatment strategies. Kidney Int. 2000;
Krishnasami Z. at al. Management of hemodialysis catheter-related bacteremia with an adjunctive antibiotic lock solution. Kidney Int. 2002)
A brief comment is required about this surprisingly difference compared to what is reported in the literature.
7- What about Mupirocin nasal use for decolonization of S. aureus in HD patients, able to reduce the rate bacteremia by 78%? is it part of the protocol?
8- The nurses’ staff attack/detachment mode was with one or two nurses (or two health workers) of support, in order to ensure greater sterility criteria during the maneuvers? In our experience only this measure alone has dramatically reduced the incidence of CVC-related infections
9- What about the baseline renal diagnosis of ESKF? Was there a concomitant immunosuppressive therapy?
10- Was there any patient on hemodialysis with previous renal transplant, to be considered more immunosuppressed?
11- What about of fungal and multiple organisms (in literature present in 7–21% of cases) and not reported by the authors?
Author Response

(The authors gave the same response as above.)

Reviewer 3 Report
Comments and Suggestions for Authors
Thank you for the opportunity to review this article. The relevance and practical significance of the study are beyond doubt since the implementation of the dialysis events prevention bundle can save the lives and health of patients. I have several questions: 1) It would be interesting to see the structure of comorbidities in the patients studied; 2) this study cannot be classified as a cohort study by design, since a cohort study does not involve any interventions, whereas the implementation of the dialysis events prevention bundle is such an intervention; 3) Figure 4 and Table 5 should provide statistically significant values.
Author Response

(The authors gave the same response as above.)

Round 2
Reviewer 1 Report
Comments and Suggestions for Authors
Dear authors
The new version of your paper is significantly improved.
I still think that the number of patients is too low, and the follow-up time is too short and that a multicenter version with longer FU would have a more important impact. Nevertheless, this time I am for the publication of your paper in the present form.
Best regards
Author Response
Dear Reviewer
Thank you for your great support in improving our manuscript. We will work on a multicenter version with a longer FU in future research.
Reviewer 3 Report
Comments and Suggestions for Authors
I am satisfied with the changes made to the manuscript by the authors. YaF recommends accepting the article for publication as it is.
Author Response
Dear Reviewer
Thank you for your great support in improving our manuscript.